# UNCERTAINTY IN MULTITASK TRANSFER LEARNING

## ABSTRACT

Using variational Bayes with neural networks, we develop an algorithm capable of accumulating knowledge into a prior from multiple different tasks. This results in a rich prior capable of few-shot learning on new tasks. The posterior can go beyond the mean field approximation and yields good uncertainty on the performed experiments. Analysis on toy tasks show that it can learn from significantly different tasks while finding similarities among them. Experiments on Mini-Imagenet reach state of the art with 74.5% accuracy on 5 shot learning. Finally, we provide two new benchmarks, each showing a failure mode of existing meta learning algorithms such as MAML and prototypical Networks.

## 1 INTRODUCTION

Recently, significant progress has been made to scale Bayesian neural networks to large tasks and to provide better approximations of the posterior distribution (Blundell et al., 2015). Recent works extend fully factorized posterior distributions to more general families (Louizos and Welling, 2017; Krueger et al., 2017; Sun et al., 2017). It is also possible to sample from the posterior distribution trough mini-batch updates (Mandt et al., 2017; Zhang et al., 2017).

However, for neural networks, the prior is often chosen for convenience. This may become a problem when the number of observations is insufficient to overcome the choice of the prior. In this regime, the prior must express our current knowledge on the task and, most importantly, our lack of knowledge on it. In addition to that, a good approximation of the posterior under the small sample size regime is required, including the ability to model multiple modes. This is indeed the case for Bayesian optimization (Snoek et al., 2012), Bayesian active learning (Gal et al., 2017), continual learning (Kirkpatrick et al., 2017), safe reinforcement learning (Berkenkamp et al., 2017), exploration-exploitation trade-off in reinforcement learning (Houthooft et al., 2016). Gaussian processes (Rasmussen, 2004) have historically been used for these applications, but an RBF kernel constitute a prior that is unsuited for many tasks. More recent tools such as deep Gaussian processes (Damianou and Lawrence, 2013) show great potential and yet their scalability whilst learning from multiple tasks needs to be improved.

Our contributions are as follow:

1. We provide a simple and scalable procedure to learn an expressive prior and posterior over models from multiple tasks.

2. We reach state of the art performances on mini-imagenet.

3. We propose two new benchmarks, each exposing a failure mode of popular meta learning algorithms. In contrast, our method perform well on these benchmarks.

   - MAML (Finn et al., 2017) does not perform well on a collection of sinus tasks when the frequency varies.
   - Prototypical Network (Snell et al., 2017)'s performance decrease considerably when the diversity of tasks increases.

**Outline:** We first describe the proposed approach in Section 2. In Section 3, we extend to three level of hierarchies and obtain a model more suited for classification. Section 4 review related methods and outline the key differences. Finally, In Section 5, we conduct experiments on three different benchmarks to gain insight in the behavior of our algorithm.

## 2 LEARNING A DEEP PRIOR

By leveraging the variational Bayesian approach, we show how we can learn a prior over models with neural networks. We start our analysis with the goal of learning a prior $p(w|\alpha)$ over the weights $w$ of neural networks across multiple tasks. We then provide a reduction of the Evidence Lower BOund (ELBO) showing that it is not necessary to explicitly model a distribution in the very high dimension of the weight space of neural networks. Instead the algorithm learns a subspace suitable for expressing model uncertainty within the distributions of tasks considered in the multi-task environment. This simplification results in a scalable algorithm which we refer to as *deep prior*.

### 2.1 HIERARCHICAL BAYES

To learn a probability distribution $p(w|\alpha)$ over the weights $w$ of a network parameterized by $\alpha$, we use a hierarchical Bayes approach across $N$ tasks, with hyper-prior $p(\alpha)$. Each task has its own parameters $w_j$, with $\mathcal{W} = \{w_j\}_{j=1}^N$. Using all datasets $\mathcal{D} = \{S_j\}_{j=1}^N$, we have the following posterior:[1]

$$p(\mathcal{W}, \alpha | \mathcal{D}) = p(\alpha | \mathcal{D}) \prod_j p(w_j | \alpha, S_j)$$

$$\propto p(\mathcal{D} | \mathcal{W}) p(\mathcal{W} | \alpha) p(\alpha)$$

$$= \prod_j \prod_i p(y_{ij} | x_{ij}, w_j) p(w_j | \alpha) p(\alpha),$$

The term $p(y_{ij}|x_{ij}, w_j)$ corresponds to the likelihood of sample $i$ of task $j$ given a model parameterized by $w_j$ *e.g.* the probability of class $y_{ij}$ from the softmax of a neural network parameterized by $w_j$ with input $x_{ij}$. For the posterior $p(\alpha|\mathcal{D})$, we assume that the large amount of data available across multiple tasks will be enough to overcome a generic prior $p(\alpha)$, such as an isotropic Normal distribution. Hence, we consider a point estimate of the posterior $p(\alpha|\mathcal{D})$ using maximum a *posteriori*[2].

We can now focus on the remaining term: $p(w_j|\alpha)$. Since $w_j$ is potentially high dimensional with intricate correlations among the different dimensions, we cannot use a simple Gaussian distribution. Following inspiration from generative models such as GANs (Goodfellow et al., 2014) and VAE (Kingma and Welling, 2013), we use an auxiliary variable $\boldsymbol{z} \sim \mathcal{N}(0, I_{d_z})$ and a deterministic function projecting the noise $\boldsymbol{z}$ to the space of $w$ *i.e.* $w = h_\alpha(\boldsymbol{z})$. Marginalizing $\boldsymbol{z}$, we have: $p(w|\alpha) = \int_{\boldsymbol{z}} p(\boldsymbol{z}) p(w|\boldsymbol{z}, \alpha) d\boldsymbol{z} = \int_{\boldsymbol{z}} p(\boldsymbol{z}) \delta_{h_\alpha(\boldsymbol{z}) - w} d\boldsymbol{z}$, where $\delta$ is the Dirac delta function. Unfortunately, directly marginalizing $\boldsymbol{z}$ is intractable for general $h_\alpha$. To overcome this issue, we add $\boldsymbol{z}$ to the joint inference and marginalize it at inference time. Considering the point estimation of $\alpha$, the full posterior is factorized as follows:

$$\prod_{j=1}^N p(w_j, \boldsymbol{z}_j | \alpha, S_j) \tag{1}$$

$$= \prod_{j=1}^N p(w_j | \boldsymbol{z}_j, \alpha, S_j) p(\boldsymbol{z}_j | \alpha, S_j)$$

$$\propto \prod_{j=1}^N p(w_j | \boldsymbol{z}_j, \alpha) p(\boldsymbol{z}_j) \prod_{i=1}^{n_j} p(y_{ij} | x_{ij}, w_j),$$

where $p(y_{ij}|x_{ij}, w_j)$ is the conventional likelihood function of a neural network with weight matrices generated from the function $h_\alpha$ i.e.: $w_j = h_\alpha(\boldsymbol{z}_j)$. Similar architecture has been used in Krueger et al. (2017) and Louizos and Welling (2017), but we will soon show that it can be reduced to a simpler architecture in the context of multi-task learning. The other terms are defined as follows:

$$p(\boldsymbol{z}_j) = \mathcal{N}(0, I) \tag{2}$$

$$p(\boldsymbol{z}_j, w_j | \alpha) = p(\boldsymbol{z}_j) \delta_{h_\alpha(\boldsymbol{z}_j) - w_j} \tag{3}$$

$$p(\boldsymbol{z}_j, w_j | \alpha, S_j) = p(\boldsymbol{z}_j | \alpha, S_j) \delta_{h_\alpha(\boldsymbol{z}_j) - w_j} \tag{4}$$

The task will consist of jointly learning a function $h_\alpha$ common to all tasks and a posterior distribution $p(\boldsymbol{z}_j|\alpha, S_j)$ for each task. At inference time, predictions are performed by marginalizing $z$ *i.e.*: $p(y|x, \mathcal{D}) = \mathbb{E}_{\boldsymbol{z}_j \sim p(\boldsymbol{z}_j|\alpha, S_j)} p(y|x, h_\alpha(\boldsymbol{z}_j))$.

---

[1] $p(x_{ij})$ cancelled with itself from the denominator since it does not depend on $w_j$ nor $\alpha$. This would have been different for a generative approach.

[2] This can be done through simply minimizing the cross entropy of a neural network with $L_2$ regularization.

## 2.2 REDUCTION OF THE EVIDENCE LOWER BOUND.

In the previous section, we described the different components for expressing the posterior distribution of Equation 4. While all these components are tractable, the normalization factor is still intractable. To address this issue, we follow the Variational Bayes approach (Blundell et al., 2015).

Conditioning on $\alpha$, we saw in Equation 1 that the posterior factorizes independently for all tasks. This reduces the joint Evidence Lower BOund (ELBO) to a sum of individual ELBO for each task.

Given a family of distributions $q_{\theta_j}(\boldsymbol{z}_j|S_j, \alpha)$, parameterized by $\{\theta_j\}_{j=1}^N$ and $\alpha$, the Evidence Lower Bound for task $j$ is:

$$
\begin{aligned}
\ln p(S_j) &\geq \underset{q(\boldsymbol{z}_j, w_j|S_j, \alpha)}{\mathbb{E}} \sum_{i=1}^{n_j} \ln p(y_{ij}|x_{ij}, w_j) - \mathrm{KL}_j \\
&= \underset{q_{\theta_j}(\boldsymbol{z}_j|S_j, \alpha)}{\mathbb{E}} \sum_{i=1}^{n_j} \ln p(y_{ij}|x_{ij}, h_\alpha(\boldsymbol{z}_j)) - \mathrm{KL}_j \\
&= \mathrm{ELBO_j},
\end{aligned}
\tag{5}
$$

where,

$$
\begin{aligned}
\mathrm{KL}_j &= \mathrm{KL}\left[q(\boldsymbol{z}_j, w_j|S_j, \alpha) \,\|\, p(\boldsymbol{z}_j, w_j|\alpha)\right] \\
&= \underset{q_{\theta_j}(\boldsymbol{z}_j|S_j, \alpha)q(w_j|\boldsymbol{z}_j, \alpha)}{\mathbb{E}} \ln \frac{q_{\theta_j}(\boldsymbol{z}_j|S_j, \alpha)}{p(\boldsymbol{z}_j|\alpha)} \frac{\delta_{h_\alpha(\boldsymbol{z}_j)-w_j}}{\delta_{h_\alpha(\boldsymbol{z}_j)-w_j}} \\
&= \underset{q_{\theta_j}(\boldsymbol{z}_j|S_j, \alpha)}{\mathbb{E}} \ln \frac{q_{\theta_j}(\boldsymbol{z}_j|S_j, \alpha)}{p(\boldsymbol{z}_j|\alpha)} \\
&= \mathrm{KL}\left[q_{\theta_j}(\boldsymbol{z}_j|S_j, \alpha) \,\|\, p(\boldsymbol{z}_j|\alpha)\right]
\end{aligned}
\tag{6}
$$

Notice that after simplification[3], $\mathrm{KL}_j$ is no longer over the space of $w_j$ but only over the space $\boldsymbol{z}_j$. Namely, the posterior distribution is factored into two components, one that is task specific and one that is task agnostic and can be shared with the prior. This amounts to finding a low dimensional manifold in the parameter space where the different tasks can be distinguished. Then, the posterior $p(\boldsymbol{z}_j|S_j, \alpha)$ only has to model which of the possible tasks are likely, given observations $S_j$ instead of modeling the high dimensional $p(w_j|S_j, \alpha)$.

But, most importantly, any explicit reference to $w$ has now vanished from both Equation 5 and Equation 6. This simplification has an important positive impact on the scalability of the proposed approach. Since we no longer need to explicitly calculate the KL on the space of $w$, we can simplify the likelihood function to $p(y_{ij}|x_{ij}, \boldsymbol{z}_j, \alpha)$, which can be a deep network parameterized by $\alpha$, taking both $x_{ij}$ and $\boldsymbol{z}_j$ as inputs. This contrasts with the previous formulation, where $h_\alpha(\boldsymbol{z}_j)$ produces all the weights of a network, yielding an extremely high dimensional representation and slow training.

## 2.3 POSTERIOR DISTRIBUTION

For modeling $q_{\theta_j}(\boldsymbol{z}_j|S_j, \alpha)$, we can use $\mathcal{N}(\boldsymbol{\mu}_j, \boldsymbol{\sigma}_j)$, where $\boldsymbol{\mu}_j$ and $\boldsymbol{\sigma}_j$ can be learned individually for each task. This, however limits the posterior family to express a single mode. For more flexibility, we also explore the usage of more expressive posterior, such as Inverse Autoregressive Flow (IAF) (Kingma et al., 2016) or Neural Autoregressive Flow (Huang et al., 2018). This gives a flexible tool for learning a rich variety of multivariate distributions. In principle, we can use a different IAF for each task, but for memory and computational reasons, we use a single IAF for all tasks and we condition[4] on an additional task specific context $\boldsymbol{c}_j$.

Note that with IAF, we cannot evaluate $q_{\theta_j}(\boldsymbol{z}_j|S_j, \alpha)$ for any values of $\boldsymbol{z}$ efficiently, only for these which we just sampled, but this is sufficient for estimating the KL term with a Monte-Carlo approxi-

---

[3]We can justify the cancellation of the Dirac delta functions by instead considering a Gaussian with finite variance, $\epsilon$. For all $\epsilon > 0$, the cancellation is valid, so letting $\epsilon \to 0$, we recover the result.

[4]We follow the architecture proposed in Kingma et al. (2016).

mation *i.e.*:

$$\text{KL}_j \approx \frac{1}{n_{\text{mc}}} \sum_{i=1}^{n_{\text{mc}}} \ln q_{\theta_j}(\boldsymbol{z}_j^{(i)}|S_j, \alpha) - \ln \mathcal{N}\left(\boldsymbol{z}_j^{(i)} \Big| \boldsymbol{0}, \boldsymbol{1}\right),$$

where $\boldsymbol{z}_j^{(i)} \sim q_{\theta_j}(\boldsymbol{z}_j|S_j, \alpha)$. It is common to approximate $\text{KL}_j$ with a single sample and let the mini-batch average the noise incurred on the gradient. We experimented with $n_{\text{mc}} = 10$, but this did not significantly improve the rate of convergence.

## 2.4 TRAINING PROCEDURE

In order to compute the loss proposed in Equation 5, we would need to evaluate every sample of every task. To accelerate the training, we use a Monte-Carlo approximation as is commonly done through the *mini-batch* procedure. First we replace summations with expectations:

$$\text{ELBO} = \sum_{j=1}^{N} \left( \mathop{\mathbb{E}}_{\boldsymbol{z}_j \sim q_j} \sum_{i=1}^{n_j} \ln p(y_{ij}|x_{ij}, z_j) - \text{KL}_j \right)$$

$$= \mathop{\mathbb{E}}_{j \sim U_N} N \left( n_j \mathop{\mathbb{E}}_{\boldsymbol{z}_j \sim q_j} \mathop{\mathbb{E}}_{i \sim U_{n_j}} \ln p(y_{ij}|x_{ij}, z_j) - \text{KL}_j \right) \tag{7}$$

Now it suffices to approximate the gradient with $n_{\text{mb}}$ samples across all tasks. Thus, we simply concatenate all datasets into a *meta-dataset* and added $j$ as an extra field. Then, we sample uniformly[5] $n_{\text{mb}}$ times with replacement from the meta-dataset. Notice the term $n_j$ appearing in front of the likelihood in Equation 7, this indicates that individually for each task it finds the appropriate trade-off between the prior and the observations. Refer to Algorithm 1 for more details on the procedure.

1:  for i in 1 .. $n_{\text{mb}}$:
2:     sample $x$, $y$ and $j$ uniformly from the meta dataset
3:     $\boldsymbol{z}_j, \ln q(\boldsymbol{z}_j) = \text{IAF}_\alpha(\boldsymbol{\mu}_j, \boldsymbol{\sigma}_j, \boldsymbol{c}_j)$
4:     $\text{KL}_j \approx \ln q(\boldsymbol{z}_j) - \ln \mathcal{N}(\boldsymbol{z}_j|0, I_{d_z})$
5:     $\mathcal{L}_i = n_j \ln p(y|x, \boldsymbol{z}_j, \alpha) + KL_j$
**Algorithm 1:** Calculating the loss for a mini-batch

## 3 EXTENDING TO 3 LEVEL OF HIERARCHIES

Deep prior gives rise to a very flexible way to transfer knowledge from multiple tasks. However, there is still an important assumption at the heart of deep prior (and other VAE-based approach such as Edwards and Storkey (2016)): the task information must be encoded in a low dimensional variable $\boldsymbol{z}$. In Section 5, we show that it is appropriate for regression, but for image classification, it is not the most natural assumption. Hence, we propose to extend to a third level of hierarchy by introducing a latent classifier on the obtained representation. This provides a simple way to enhance existing algorithm such as Prototypical Networks (Proto Net) (Snell et al., 2017).

In Equation 5, for a given[6] task $j$, we decomposed the likelihood $p(S|z)$ into $\prod_{i=1}^n p(y_i|x_i, z)$ by assuming that the neural network is directly predicting $p(y_i|x_i, z)$. Here, we introduce a latent variable $v$ to make the prediction $p(y_i|x_i, v)$. This can be, for example, a Gaussian linear regression on the representation $\phi_\alpha(x, \boldsymbol{z})$ produced by the neural network. The general form now factorizes as follow: $p(S|z) = \mathop{\mathbb{E}}_{v \sim p(v|z)} \prod_i p(y_i|v, x_i)p(x_i)$, which is commonly called the *marginal likelihood*.

To compute $\text{ELBO}_j$ in 5 and update the parameters $\alpha$, the only requirement is to be able to compute the marginal likelihood $p(S|z)$. There are closed form solutions for, e.g., linear regression with Gaussian prior, but our aim is to compare with algorithms such as Prototypical Networks on a

---

[5]We also explored a sampling scheme that always make sure to have at least $k$ samples from the same task. The aim was to reduce gradient variance on task specific parameters but, we did not observed any benefits.

[6]We removed $j$ from equations to alleviate the notation.

classification benchmark. Alternatively, we can factor the marginal likelihood as follow $p(S|z) = \prod_{i=1}^{n} p(y_i|x_i, S_{0..i-1}, z)$. If a well calibrated task uncertainty is not required, one can also use a leave-one-out procedure $\prod_{i=1}^{n} p(y_i|x_i, S \setminus \{x_i, y_i\}, z)$. Both of these factorizations correspond to training $n$ times the latent classifier on a subset of the training set and evaluating on a sample left out. We refer the reader to Rasmussen (2004, Chapter 5) for a discussion on the difference between leave-one-out cross-validation and marginal likelihood.

For a practical algorithm, we propose a closed form solution for leave-one-out in prototypical networks. In its standard form, the prototypical network produces a prototype $c_k$ by averaging all representations $\gamma_i = \phi_\alpha(x_i, z)$ of class $k$ *i.e.* $c_k = \frac{1}{|K|} \sum_{i \in K} \gamma_i$, where $K = \{i : y_i = k\}$. Then, predictions are made using $p(y = k|x, \alpha, z) \propto \exp\left(-\|c_k - \gamma_i\|_2\right)$.

**Theorem 1.** *Let $c_k^{-i} \ \forall k$ be the prototypes computed without example $x_i, y_i$ in the training set. Then,*

$$\|c_k^{-i} - \gamma_i\|_2 = \begin{cases} \frac{|K|}{|K|-1}\|c_k - \gamma_i\|_2, & \text{if } y_i = k \\ \|c_k - \gamma_i\|_2, & \text{otherwise} \end{cases} \tag{8}$$

We defer the proof to supplementary materials. Hence, we only need to compute prototypes once and rescale the Euclidean distance when comparing with a sample that was used for computing the current prototype. This gives an efficient algorithm with the same complexity as the original one and a good proxy for the marginal likelihood.

## 4 RELATED WORK

Hierarchical Bayes algorithms for multitask learning has a long history (Daumé III, 2009; Wan et al., 2012; Bakker and Heskes, 2003). However most of the literature focuses on simple statistical models and does not consider transferring on new tasks.

More recently, Edwards and Storkey (2016) and Bouchacourt et al. (2017) explore hierarchical Bayesian inference with neural networks and evaluate on new tasks. Both papers use a two-level Hierarchical VAE for modeling the observations. While similar, our approach differs in a few different ways. We use a discriminative approach and focus on model uncertainty. We show that we can obtain a posterior on $z$ without having to explicitly encode $S_j$. We also explore the usage of more complex posterior family such as IAF. these differences make our algorithm simpler to implement, and easier to scale to larger datasets. Other works consider neural networks with latent variables (Tang and Salakhutdinov, 2013; Depeweg et al., 2017; Turner et al., 2018) but does not explore the ability to learn across multiple tasks.

Some recent works on meta-learning are also targeting transfer learning from multiple tasks. Model-Agnostic Meta-Learning (MAML) (Finn et al., 2017) finds a shared parameter $\theta$ such that for a given task, one gradient step on $\theta$ using the training set will yield a model with good predictions on the test set. Then, a meta-gradient update is performed from the test error through the one gradient step in the training set, to update $\theta$. This yields a simple and scalable procedure which learns to generalize. Recently Grant et al. (2018) considers a Bayesian version of MAML. Additionally, (Ravi and Larochelle, 2016) also consider a meta-learning approach where an encoding network reads the training set and generates the parameters of a model, which is trained to perform well on the test set.

Finally, some recent interest in few-shot learning give rise to various algorithms capable of transferring from multiple tasks. Many of these approaches (Vinyals et al., 2016; Snell et al., 2017) find a representation where a simple algorithm can produce a classifier from a small training set. Bauer et al. (2017) use a neural network pre-trained on a standard multi-class dataset to obtain a good representation and use classes statistics to transfer prior knowledge to new classes.

## 5 EXPERIMENTAL RESULTS

Through experiments, we want to answer i) Can deep prior learn a meaningful prior on tasks? ii) Can it compete against state of the art on a strong benchmark? iii) In which situations does deep prior and other approaches fail?

## 5.1 REGRESSION ON ONE DIMENSIONAL HARMONIC SIGNALS

To gain a good insight into the behavior of the prior and posterior, we choose a collection of one dimensional regression tasks. We also want to test the ability of the method to *learn* the task and not just match the observed points. For this, we will use periodic functions and test the ability of the regressor to extrapolate outside of its domain.

Specifically, each dataset consists of $(x, y)$ pairs (noisily) sampled from a sum of two sine waves with different phase and amplitude and a frequency ratio of 2: $f(x) = a_1 \sin(\omega \cdot x + b_1) + a_2 \sin(2 \cdot \omega \cdot x + b_2)$, where $y \sim \mathcal{N}(f(x), \sigma_y^2)$. We construct a meta-training set of 5000 tasks, sampling $\omega \sim \mathcal{U}(5, 7)$, $(b_1, b_2) \sim \mathcal{U}(0, 2\pi)^2$ and $(a_1, a_2) \sim \mathcal{N}(0, 1)^2$ independently for each task. To evaluate the ability to extrapolate outside of the task's domain, we make sure that each task has a different domain. Specifically, $x$ values are sampled according to $\mathcal{N}(\mu_x, 1)$, where $\mu_x$ is sample from the *meta-domain* $\mathcal{U}(-4, 4)$. The number of training samples ranges from 4 to 50 for each task and, evaluation is performed on 100 samples from tasks never seen during training.

**Model**  Once $z$ is sampled from IAF, we simply concatenate it with $x$ and use 12 densely connected layers of 128 neurons with residual connections between every other layer. The final layer linearly projects to 2 outputs $\mu_y$ and $s$, where $s$ is used to produce a heteroskedastic noise, $\sigma_y = \text{sigmoid}(s) \cdot 0.1 + 0.001$. Finally, we use $p(y|x, z) = \mathcal{N}(\mu_y(x, z), \sigma_y(x, z)^2)$ to express the likelihood of the training set. To help gradient flow, we use ReLU activation functions and Layer Normalization[7] (Ba et al., 2016).

**Results**  Figure 1a depicts examples of tasks with 1, 2, 8, and 64 samples. The true underlying function is in blue while 10 samples from the posterior distributions are faded in the background. The thickness of the line represent 2 standard deviations. The first plot has only one single data point and mostly represents samples from the prior, passing near this observed point. Interestingly, all samples are close to some parametrization of Equation 5.1. Next with only 2 points, the posterior is starting to predict curves highly correlated with the true function. However, note that the uncertainty is over optimistic and that the posterior failed to fully represent all possible harmonics fitting these two points. We discuss this issue more in depth in supplementary materials. Next, with 8 points, it managed to mostly capture the task, with reasonable uncertainty. Finally, with 64 points the model is certain of the task.

To add a strong baseline, we experimented with MAML (Finn et al., 2017). After exploring a variety of values for hyper-parameter and architecture design we couldn't make it work for our two harmonics meta-task. We thus reduced the meta-task to a single harmonic and reduced the base frequency range by a factor of two. With these simplifications, we managed to make it converge, but the results are far behind that of deep prior even in this simplified setup. Figure 1b shows some form of adaptation with 16 samples per task but the result is jittery and the extrapolation capacity is very limited. these results were obtained with a densely connected network of 8 hidden layers of 64 units[8], with residual connections every other layer. The training is performed with two gradient steps and the evaluation with 5 steps. To make sure our implementation is valid, we first replicated their regression result with a fixed frequency as reported in (Finn et al., 2017).

Finally, to provide a stronger baseline, we remove the KL regularizer of deep prior and reduced the posterior $q_{\theta_j}(z_j|S_j, \alpha)$ to a deterministic distribution centered on $\mu_j$. The mean square error is reported in Figure 2 for an increasing dataset size. This highlights how the uncertainty provided by deep prior yields a systematic improvement.

## 5.2 MINI-IMAGENET EXPERIMENT

Vinyals et al. (2016) proposed to use a subset of Imagenet to generate a benchmark for few-shot learning. Each task is generated by sampling 5 classes uniformly and 5 training samples per class, the remaining images from the 5 classes are used as query images to compute accuracy. The number of unique classes sums to 100, each having 600 examples of $84 \times 84$ images. To perform meta-validation and meta-test on unseen tasks (and classes), we isolate 16 and 20 classes respectively from the original

---

[7]Layer norm only marginally helped.

[8]We also experimented with various other architectures.

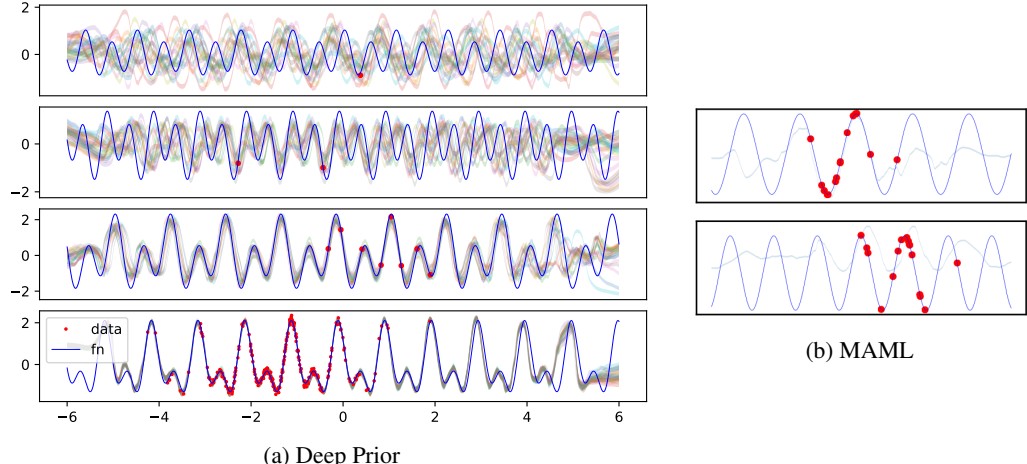

(b) MAML

(a) Deep Prior

Figure 1: Preview of a few tasks (blue line) with increasing amount of training samples (red dots). Samples from the posterior distribution are shown in semi-transparent colors. The width of each samples is two standard deviations (provided by the predicted heteroskedastic noise).

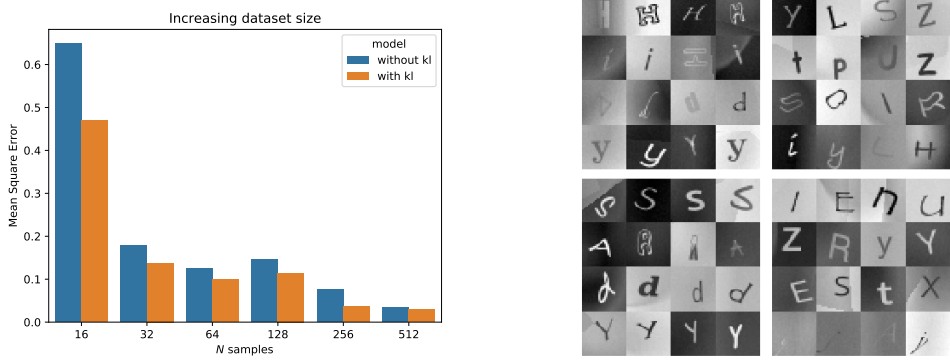

Figure 2: **left:** Mean Square Error on increasing dataset size. The baseline corresponds to the same model without the KL regularizer. Each value is averaged over 100 tasks and 10 different restart. **right:** 4 sample tasks from the Synbols dataset. Each row is a class and each column is a sample from the classes. In the 2 left tasks, the symbol have to be predicted while in the two right tasks, the font has to be predicted.

set of 100, leaving 64 classes for the training tasks. This follows the procedure suggested in Ravi and Larochelle (2016).

The training procedure proposed in Section 2 requires training on a fixed set of tasks. We found that 1000 tasks yields enough diversity and that over 9000 tasks, the embeddings are not being visited often enough over the course of the training. To increase diversity during training, the $5 \times 5$ training and test sets are re-sampled every time from a fixed train-test split of the given task[9].

We first experimented with the vanilla version of deep prior (2). In this formulation, we use a ResNet (He et al., 2016) network, where we inserted FILM layers (Perez et al., 2017; de Vries et al., 2017) between each residual block to condition on the task. Then, after flattening the output of the final convolution layer and reducing to 64 hidden units, we apply a $64 \times 5$ matrix generated from a transformation of $z$. Finally, predictions are made through a softmax layer. We found this architecture to be slow to train as the generated last layer is noisy for a long time and prevent the rest of the

---

[9]If the train and test split is not fixed for a given task, one could leak the test information through the task embeddings across different resampling of the task.

| | Accuracy |
|---|---|
| Matching Networks (Vinyals et al., 2016) | 60.0 % |
| Meta-Learner (Ravi and Larochelle, 2016) | 60.6 % |
| MAML (Finn et al., 2017) | 63.2% |
| Prototypical Networks (Snell et al., 2017) | 68.2 % |
| SNAIL (Mishra et al., 2018) | 68.9 % |
| Discriminative k-shot (Bauer et al., 2017) | 73.9 % |
| adaResNet (Munkhdalai et al., 2018) | 71.9 % |
| Deep Prior (Ours) | 62.7 % |
| **Deep Prior + Proto Net (Ours)** | **74.5 %** |

Table 1: Average classification accuracy on 5-shot Mini-Imagenet benchmark.

| | 5-way, 5-shot Mini-Imagenet | 4-way, 4-shot Synbols |
|---|---|---|
| Proto Net (ours) | $68.6 \pm 0.5\%$ | $69.6 \pm 0.8\%$ |
| + ResNet(12) | $72.4 \pm 1.0\%$ | $76.8 \pm 0.4\%$ |
| + Conditioning | $72.3 \pm 0.6\%$ | $80.1 \pm 0.9\%$ |
| + Leave-One-Out | $73.9 \pm 0.4\%$ | $82.7 \pm 0.2\%$ |
| + KL | $\mathbf{74.5 \pm 0.5\%}$ | $\mathbf{83.5 \pm 0.4\%}$ |

Table 2: Ablation Study of our model. Accuracy is shown with 90% confidence interval over bootstrap of the validation set.

network to learn. Nevertheless, we obtained 62.6% accuracy on Mini-Imagenet, on par with many strong baselines.

To enhance the model, we combine task conditioning with prototypical networks as proposed in Section 3. This approach alleviates the need to generate the final layer of the network, thus accelerating training and increasing generalization performances. While we no longer have a well calibrated task uncertainty, the KL term still acts as an effective regularizer and prevents overfitting on small datasets[10]. With this improvement, we are now the new state of the art with 74.5% (Table 1). In Table 2, we perform an ablation study to highlight the contributions of the different components of the model. In sum, a deeper network with residual connections yields major improvements. Also, task conditioning does not yield improvement if the leave-one-out procedure is not used. Finally, the KL regularizer is the final touch to obtain state of the art.

## 5.3 HETEROGENEOUS COLLECTION OF TASKS

In Section 5.2, we saw that conditioning helps, but only yields a minor improvement. This is due to the fact that Mini-Imagenet is a very homogeneous collection of tasks where a single representation is sufficient to obtain good results. To support this claim, we provide a new benchmark[11] of synthetic symbols which we refer to as Synbols. Images are generated using various font family on different alphabets (Latin, Greek, Cyrillic, Chinese) and background noise (Figure 2, right). For each task we have to predict either a subset of 4 font families or 4 symbols with only 4 examples. Predicting either fonts or symbols with two separate Prototypical Networks, yields 84.2% and 92.3% accuracy respectively, with an average of 88.3%. However, blending the two collections of tasks in a single benchmark, brings prototypical network down to 76.8%. Now, conditioning on the task with deep prior brings back the accuracy to 83.5%. While there is still room for improvement, this supports the claim that a single representation will only work on homogeneous collection of tasks and that task conditioning helps learning a family of representations suitable for heterogeneous benchmarks.

## 6 CONCLUSION

Using a variational Bayes framework, we developed a scalable algorithm for hierarchical Bayesian learning of neural networks, called deep prior. This algorithm is capable of transferring information from tasks that are potentially remarkably different. Results on the Harmonics dataset shows that the learned manifold across tasks exhibits the properties of a meaningful prior. Finally, we found that MAML, while very general, will have a hard time adapting when tasks are too different. Also, we found that algorithms based on a single image representation only works well when all tasks can succeed with a very similar set of features. Together, these findings allowed us to reach the state of the art on Mini-Imagenet.

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

# 7 APPENDIX

## 7.1 PROOF OF LEAVE-ONE-OUT

**Theorem 1.** *Let $c_k^{-i}$ $\forall k$ be the prototypes computed without example $x_i, y_i$ in the training set. Then,*

$$\|c_k^{-i} - \phi_\alpha(x_i)\|_2 = \begin{cases} \frac{|K|}{|K|-1}\|c_k - \phi_\alpha(x_i)\|_2, & \text{if } y_i = k \\ \|c_k - \phi_\alpha(x_i)\|_2, & \text{otherwise} \end{cases} \tag{9}$$

*Proof.* Let $\gamma_i = \phi_\alpha(x_i)$, $n = |K|$ and assume $y_i = k$ then,

$$\gamma_i - c_k^{-i} = \gamma_i - \frac{1}{n-1} \sum_{j \in K \wedge j \neq i} \gamma_j \tag{10}$$

$$= \gamma_i - \frac{1}{n-1} \left( \sum_{j \in K \wedge j \neq i} \gamma_j + \gamma_i - \gamma_i \right) \frac{n-1}{n} \frac{n}{n-1} \tag{11}$$

$$= \gamma_i \left( 1 + \frac{1}{n-1} \right) - \frac{n}{n-1} \left( \frac{1}{n} \sum_{j \in K} \gamma_j \right) \tag{12}$$

$$= \frac{n}{n-1} \left( \gamma_i - c_k \right). \tag{13}$$

When $y_i \neq k$, the result is trivially $\gamma_i - c_k^{-i} = \gamma_i - c_k$. $\qquad\square$

## 7.2 LIMITATIONS OF IAF

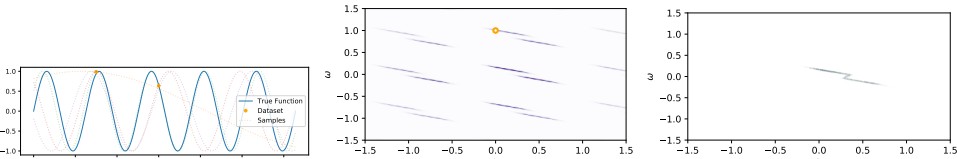

Figure 3: **top:** True function in the original space with 2 observed data points. **middle:** True posterior distribution, where the orange dot corresponds to the location of the true underlying function. **bottom:** Samples from IAF's learned posterior.

When experimenting with the Harmonics toy dataset in Section 5.1, we observed issues with repeatability, most likely due to local minima. We decided to investigate further on the multimodality of posterior distributions with small sample size and the capacity of IAF to model them. For this purpose we simplified the problem to a single sine function and removed the burden of learning the prior. The likelihood of the observations is defined as follows:

$$f(x) = \sin(5(\omega \cdot x + b)); \quad y \sim \mathcal{N}(f(x), \sigma_y^2),$$

where $\sigma_y = 0.1$ is given and $p(\omega) = p(b) = \mathcal{N}(0, 1)$. Only the frequency $\omega$ and the bias $b$ are unknown[12], yielding a bi-dimensional problem that is easy to visualize and quick to train. We use a dataset of 2 points at $x = 1.5$ and $x = 3$ and the corresponding posterior distribution is depicted in Figure 3-middle, with an orange point at the location of the true underlying function. Some samples from the posterior distribution can be observed in Figure 3-top.

We observe a high amount of multi-modality on the posterior distribution (Figure 3-middle). Some of the modes are just the mirror of another mode and correspond to the same functions e.g. $b + 2\pi$ or $-f$ ; $b + \pi$. But most of the time they correspond to different functions and modeling them is crucial for some application. The number of modes varies a lot with the choice of observed dataset, ranging from a few to several dozens. Now, the question is: "How many of those modes can IAF model?". Unfortunately, Figure 3-bottom reveals poor capability for this particular case. After carefully adjusting the hyperparameters[13] of IAF, exploring different initialization schemes and running multiple restarts, we rarely capture more than two modes (sometimes 4). Moreover, it will not be able to fully separate the two modes. There is systematically a thin path of density connecting each modes as a chain. With longer training, the path becomes thinner but never vanishes and the magnitude stays significant.

---

[12] We scale $\omega$ and $b$ by a factor of 5 so that the range of interesting values fits well in the interval $(-1, 1)$. This makes it more approachable by IAF.

[13] 12 layers with 64 hidden units MADE network for each layer, learned with Adam at a learning rate of 0.0002.

