# OpenReview forum: "Uncertainty in Multitask Transfer Learning"
_ICLR.cc/2019/Conference_

### Official Review · AnonReviewer1 · 2018-10-17
**Method that seem to work in practice, but needs better comparison and has issues with presentation**

**Rating:** 4
**Confidence:** 4

**Review:**

The paper presents a method for training a probabilistic model for Multitask Transfer Learning. The key idea is to introduce a latent variable "z" per task which to capture the commonality in the task instances. Since this leads to an intractable likelihood the authors use the standard ELBO with a Variational Distribution over "z" defined as a Gaussian + Inverse Autoregressive Flow. For classification, the authors also show that they can combine the model with the main idea in Prototypical Networks.

The experiments evaluate on three different task, the comparison against MAML on the toy problem is quite interesting. However, the results on the Mini-Imagenet suggest that the main contributors to the better performance are the Prototypical Networks idea and the improved ResNet. Additionally, the authors compare against MAML only on the toy task and not on their synthetic dataset. I think that the experiments need better comparisons (there have been published an improved version of MAML, or even just add results from your own implementation of MAML with the same ResNet on the 3rd task as well).

A major issue is that the model presented is not really a Hierarchical Bayesian model as being strongly presented. It is much more a practical variational algorithm, which is not bad by no means, but I find its "interpretation" as a Hierarchical Bayesian method as totally unnecessary and making the paper significantly harder to read and follow than it needs to be. This is true for both the base model and the model + ProtoNet. I think that the manuscript itself requires more work as well as a better comparison of the method to baseline algorithms.


Section 2.2:

The authors start by introducing a "Hierarchical Bayes" model over the parameters of a Neural Network for multi-task learning. By defining the model parameters to be an implicit function of some low-dimensional noise and the hyper-parameter they shift the inference to the noise variable "z". One issue, which I won't discuss further, is that this defines a degenerate distribution over the parameters (a fact well known in the GAN literature), which seem counter-intuitive to call "Bayesian". Later, since the parameters "w" has vanished from the equation the authors conclude that now they can change the whole graphical models such that there is actually no distribution over the parameters of a Neural Network, while the hyper-parameter IS now the parameters of a Neural Network and the latent variable is an input to it. Mathematically, the transformation is valid, however, this no longer corresponds to the original graphical model that was described earlier. The procedure described here is essentially a Variational Model with latent variable "z" for each task and the method performs a MAP estimation of the parameters of the Generative Model by doing Variational Inference (VAE to be exact) on the latent "z". There is nothing bad about this model, however, the whole point of using a "Hierarchical Bayes" for the parameters of the Network serves no purpose and is significantly different to the actual model that is proposed.

In section 2, the prior term p(a) in equation 7 and Algorithm 1 is missing.

Section 3:

The authors argue that they add yet another level of hierarchy in the Graphical Model with a further latent variable "v", which is unclear fundamentally why you need it as it can be subsumed inside "z" (from a probabilistic modelling perspective they play similar roles). Additionally, they either do not include a prior or on "v" or there is a mistake in the equation for p(S|z) at the bottom of page 4. The main motivation for this comes from the literature where for instance if we have a linear regression and "v" represents the weights of the last linear layer with a Gaussian Prior than the posterior over "v" has an analytical form. After this whole introduction into the special latent variable "v", the authors actually use the idea from Prototypical Networks. They introduce a valid leave-one producer for training. However, the connection to the latent variable "v" which was argued to be the third level of a Hierarchical Bayes model is now lost, as the context c_k is no longer a separate latent variable (it has no prior and in the original Prototypical Network although the idea can be interpreted in a probabilistic framework it is never presented as a Hierarchical Bayes).

---

> ### Author Response · Authors · 2018-11-27
> **Addressing Reviewer's Concerns**
>
> We thank the reviewer for the constructive comments.
>
> In response to : “… The main contributors to the better performance are the Prototypical Networks idea and the improved ResNet…”
>
> ResNet is indeed an important contribution to the increased performance. We employ ResNet to be competitive with recent methods e.g. Discriminative k-short learning. An ablation study is conducted to be explicit about the relative contributions. Despite this, there is still an extra 2% gained with our variational algorithm which is necessary to reach state of the art. But most importantly, experiments on “Synbols” show that the task conditioning is necessary. We believe that these results only are sufficient for a strong contribution.
>
> In response to : “Section 2.2 ...”
>
> We introduce the algorithm through a Hierarchical Bayes perspective as a way to justify our method and to be explicit on where the assumptions and approximations are made. Being principled and sound should not be considered a negative aspect. We are willing to rewrite this part to simplify the paper to make it more accessible.
>
> In response to: “Additionally, they either do not include a prior or on "v" or there is a mistake in the equation for p(S|z)”
>
> The expectation is over p(v|z). Hence, the prior over “v” is there. Following the notation of the paper, it is under the Expectation.
>
> In response to: “... with a further latent variable "v", which is unclear fundamentally why you need it as it can be subsumed inside "z"...”
>
> $z$ is a latent variable representing the whole task and $v$ encodes a given image instance. They do not contain the same information. While the notion of representation learning is powerful, the experiments on synbols shows that it lacks adaptability. Instead, we learn a “family” of representation conditioned on $z$ and we adapt the representation for the given task.

---

> > ### Comment · AnonReviewer1 · 2018-11-30
> > **Thanks for the comment**
> >
> > Following and the other reviewers I would indeed suggest the whole rewrite of section 2.2 and not talking about Hierarchical Bayes, but rather the VI framework that you actually use in practice. I do feel this is currently a major issue and needs significant addressing to make the paper publication ready.
> >
> > "Being principled and sound should not be considered a negative aspect." - I might have not explained my point well here, but it is exactly for the fact that you are talking about a hierarchical bayes over the weights of the network, which is not what you end up using. This is my main critisism, otherwise I agree principled motivation is not to be discouraged, even if you do approximate versions of it later.
> >
> > Thanks for clarifying the rolve of the latent variable v.

---

### Official Review · AnonReviewer3 · 2018-11-03
**The authors propose a generative model for multitask learning using task-specific latent variables. Unfortunately, the paper has strong technical and presentational shortcomings.**

**Rating:** 2
**Confidence:** 5

**Review:**

The authors state that their goal with this paper is manifold:
They want to learn a prior over neural networks for multiple tasks. The posterior should go beyond mean field inference and yield good results.  The authors claim in their paper that they learn an 'expressive transferable prior over the weights of a network' for multi-task settings, which they denote with the unfortunate term 'deep prior'.

In sec. 2.1 the authors introduce the idea of a hierarchical probabilistic model of weights for a neural network p(W|a) conditioned on task latent variables p(a). They realize that one might want to generate those weights with a function which conditions on variable "z" and has parameters "a". They continue their argument in Sec 2.2 that since the weight scoring can be canceled out in the ELBO, the score of the model does not depend on weights "w" explicitly anymore.
This, of course, is wrong, since the likelihood term in the ELBO still is an expectation over the posterior of q(w|z)q(z).
However, the authors also realize this and continue their argumentation as follows:
In this case -according to the authors- one may drop the entire idea about learning distributions over weights entirely.
The math says: p(y|x ; a) = int_z p(z) int_w p(w|z ; a) p(y|x, w)dw dz.
So the authors claim that a model p(y|x, z) which only conditions on 'z' is the same as the full Bayesian Model with marginalized weights. They then suggest to just use any neural network with parameters "a" to model this p(y|x, z ;a) directly with z being used as an auxiliary input variable to the network with parameters "a" and claim this is doing the same. This is of course utterly misleading, as the parameter "a" in the original model indicated a model mapping a low dimensional latent variable to weights, but now a maps to a neural network mapping a latent variable and an input vector x to an output vector y. As such, these quantities are different and the argument does not hold. Also a point estimate of said mapping will not be comparable to the marginalized p(y|x).

What is more concerning is that the authors claim this procedure is equivalent to learning a distribution over weights and call the whole thing a deep prior, while this paper contains no work on trying to perform the hard task of successfully parametrizing a high-dimensional conditional distribution over weights p(w|z) (apart from a trivial experiment generating all of them at once  from a neural network for a single layer in a failed experiment) but claims to succeed in doing so by circumventing it entirely.

In their experiments, the authors also do not actually successfully try to really learn a full distribution over the weights of a neural network. This alone suffices to realize that the paper appears to be purposefully positioned in a highly misleading way and makes claims about weight priors that are superficially discussed in various sections but never actually executed on properly in the paper.
This is a disservice to the hard work many recent and older papers are doing in actually trying to derive structured hierarchical weight distributions for deep networks, which this paper claims is a problem they find to be 'high dimensional and noisy', which is exactly why it is a valid research avenue to begin with that should not be trivially subsumed by work such as this.

When reducing this paper to the actual components it provides, it is a simple object: A deterministic neural network with an auxiliary, task-dependent latent variable which provides extra inputs to model conditional densities.
Such ideas have been around for a while and the authors do not do a good job of surveying the landscape of such networks with additional stochastic input variables.
One example is "Learning Stochastic Feedforward Neural Networks" by Tang and Salakhutdinov, NIPS 2013, a more recent one is "Uncertainty Decomposition in Bayesian Neural Networks with Latent Variables" by Depeweg et al 2017.
An obvious recent example of multi-task/meta/continual learning comparators would be "VARIATIONAL CONTINUAL LEARNING" by Nguyen et al. and other work from the Cambridge group that deals with multi-task and meta-learning and priors for neural networks.

Another weakness of the paper is that the main driver of success in the paper's experiment regarding classification is the prototypical network idea, rather than anything else regarding weight uncertainty which seems entirely disentangled from the core theoretical statements of the paper.

All in all, I find this paper unacceptably phrased with promises it simply does not even attempt to keep and a misleading technical section that would distort the machine learning literature without actually contributing to a solution to the technical problems it claims to tackle (in relation to modeling weight uncertainty/priors on NN). Paired with the apparent disinterest of the authors to cite recent and older literature executing strongly related underlying ideas combining neural networks with auxiliary latent variables, I can only recommend that the authors significantly change the writing and the attribution of ideas in this paper for a potential next submission focusing on multi-task learning and clarify and align the core ideas in the theory sections and the experiment sections.

---

> ### Author Response · Authors · 2018-11-26
> **Addressing Reviewers Concerns**
>
> This review contains false accusations and strong opinions. We kindly ask the reviewer to stay factual for the rest of this discussion.
>
> This paper was proofread by strong mathematicians. All of whom agreed that the derivation in section 2.2 is sound. If the reviewer believes that there is a mathematical error please be more precise on its location.
>
> We agree that some of the phrasing in the paper may bring confusion. What appears to be at the center of the confusion is the difference between a prior over models and a prior over weights of a neural network. After rephrasing some sentences, we made clear that the resulting algorithms learn a prior over models and not a prior over the weights of a network. More specifically, we modified the introduction of Section 2 as follow:
>
> By leveraging the variational Bayesian approach, we show how we can learn a prior over models with neural networks. We start our analysis with the goal of learning a prior $p(w|\alpha)$ over the weights $w$ of neural networks across multiple tasks. We then provide a reduction of the Evidence Lower BOund (ELBO) showing that it is not necessary to explicitly model a distribution in the very high dimension of the weight space of neural networks. Instead the algorithm learns a subspace suitable for expressing model uncertainty within the distributions of tasks considered in the multi-task environment.
>
> AnonReviewer3 wrote: “...which is exactly why it is a valid research avenue to begin with that should not be trivially subsumed by work such as this.”
>
> We do not seek to undermine the research on posterior over weight uncertainty. Our result only addresses model uncertainty in multi-task environments. Also, any progress on better posterior over weight uncertainty can be applied over $\alpha$, the weights of the “main” network. The updated version is more clear about that.
>
> An extensive review of multi-task learning, hierarchical Bayes, meta-learning, few-shot learning and weight uncertainty goes beyond the scope of this paper. We did a significant effort to cover these domains with two pages of citations. The new version now includes all recommended citations and more.

---

### Official Review · AnonReviewer2 · 2018-11-09
**The work proposes a variational approach to meta-learning that employs latent variables corresponding to task-specific datasets, but is presented in a misleading and imprecise manner. The experimental improvements are not well-motivated by the methodology introduced in the paper.**

**Rating:** 3
**Confidence:** 5

**Review:**

Strengths:
+ A variational approach to meta-learning is timely in light of recent approaches to solving meta-learning problems using a probabilistic framework.
+ The experimental result on a standard meta-learning benchmark, miniImageNet, is a significant improvement.

Weaknesses:
- The paper is motivated in a confusing manner and neglects to thoroughly review the literature on weight uncertainty in neural networks.
- The SotA result miniImageNet is the result of a bag-of-tricks approach that is not well motivated by the main methodology of the paper in Section 2.

Major points:
- The motivation for and derivation of the approach in Section 2 is misleading, as the resulting algorithm does not model uncertainty over the weights of a neural network, but instead a latent code z corresponding to the task data S. Moreover, the approach is not fully Bayesian as a point estimate of the hyperparameter \alpha is computed; instead, the approach is more similar to empirical Bayes. The submission needs significant rewriting to clarify these issues. I also suggest more thoroughly reviewing work on explicit weight uncertainty (e.g., https://arxiv.org/abs/1505.05424 , http://proceedings.mlr.press/v54/sun17b.html , https://arxiv.org/abs/1712.02390 ).
- Section 3, which motivates a combination of the variational approach and prototypical networks, is quite out-of-place and unmotivated from a probabilistic perspective. The motivation is deferred to Section 5 but this makes Section 3 quite unreadable. Why was this extraneous component introduced, besides as a way to bump performance on miniImageNet?
- The model for the sinusoidal data seems heavily overparameterized (12 layers * 128 units), and the model for the miniImageNet experiment (a ResNet) has significantly more parameters than models used in Prototypical Networks and MAML.
- The training and test set sampling procedure yields a different dataset than the one used in e.g., MAML or Prototypical Networks. Did the authors reproduce the results reported in Table 1 using their dataset?

Minor points:
- abstract: "variational Bayes neural networks" -> variational Bayesian neural networks, but also this mixes an inference procedure with just being Bayesian
- pg. 1: "but an RBF kernel constitute a prior that is too generic for many tasks" give some details as to why?
- pg. 2: "we extend to three level of hierarchies and obtain a model more suited for classification" This is not clear.
- pg. 2: " variational Bayes approach" -> variational Bayesian approach OR approach of variational Bayes
- pg. 2: "scalable algorithm, which we refer to as deep prior" This phrasing is strange to me. A prior is an object, not an algorithm, and moreover, the word "deep" is overloaded in this setting.
- pg. 3: "the normalization factor implied by the "∝" sign is still intractable." This is not good technical presentation.
- pg. 3: "we use a single IAF for all tasks and we condition on an additional task specific context cj" It might be nice to explore or mention that sharing parameters might be helpful in the multitask setting...
- Section 2.4 describes Robbins & Munro style estimation. Why call this the "mini-batch" principle?

---

> ### Author Response · Authors · 2018-11-27
> **Addressing Reviewer's Concerns**
>
> We thank the reviewer for the constructive comments.
>
> In response to : “ The motivation for and derivation of the approach in Section 2 is misleading, as the resulting algorithm does not model uncertainty over the weights of a neural network, but instead a latent code z corresponding to the task data S.”
>
> We agree that in its current form, the paper can be misleading. We rephrased a few sentences to make it clear that we seek to learn model uncertainty within the task distribution and not the weight uncertainty of an hypothetical neural network. Introduction of Section 2 is now as follow:
>
> “By leveraging the variational Bayesian approach, we show how we can learn a prior over models with neural networks. We start our analysis with the goal of learning a prior $p(w|\alpha)$ over the weights $w$ of neural networks across multiple tasks. We then provide a reduction of the Evidence Lower BOund (ELBO) showing that it is not necessary to explicitly model a distribution in the very high dimension of the weight space of neural networks. Instead the algorithm learns a subspace suitable for expressing model uncertainty within the distributions of tasks considered in the multi-task environment.”
>
> In response to : “ the approach is not fully Bayesian as a point estimate of the hyperparameter \alpha is computed...”
>
> In the first version of this work we had equations considering uncertainty on \alpha. Early experiments however showed no benefits. Hence, we decided to treat \alpha as a point estimate to make the paper more accessible. This is analogous to ignoring the uncertainty on the encoder/decoder, which is common in the VAE literature.
>
> In response to: “I also suggest more thoroughly reviewing work on explicit weight uncertainty…”
>
> The first reference was already cited in the paper. The next two are now included and we extended the literature review.
>
> In response to: “Section 3, which motivates a combination of the variational approach and prototypical networks, is quite out-of-place and unmotivated from a probabilistic perspective...”
>
> The aim is to show how to augment a practical algorithm like Proto Net and make it more adaptable. We obtain state of the art on mini-imagenet and we show on the “Synbols” benchmark an important performance gap (28% error rate reduction). The take home of this experiment is that Proto Net alone is not sufficient if we want to consider general multi task learning. We believe that the result of this experiment is highly important to lead the community in the right direction.
>
> In response to: “The model for the sinusoidal data seems heavily overparameterized… ”
>
> We compare to a version of ProtoNet and MAML with same architecture and parameters. Hence our conclusions are valid. Some competing algorithms use large networks with residual connections e.g. Discriminative k shot learning uses ResNet-34. Also, in our case, we need to blend the input x with the task representation z. Hence, to minimize the chances of underfitting, we use a high capacity network.
>
> In response to: “The training and test set sampling procedure yields a different dataset than the one used in e.g., MAML or Prototypical Networks. Did the authors reproduce the results reported in Table 1 using their dataset?”
>
> It is the same datasets with the same splits. The sampling procedure to generate a mini-batch during training is indeed different, but does not affect the evaluation procedure: a new task with classes never seen is presented with a small training and testing sets.
>
>
> In response to : “Minor points”
> Corrections were added to the document.

---

### Meta-Review · Area_Chair1 · 2018-12-16
**Presentation shortcomings**

**Confidence:** 5
**Recommendation:** Reject

**Metareview:**

This paper presents a meta-learning approach which relies on a learned prior over neural networks for different tasks.

The reviewers found this work to be well-motivated and timely. While there are some concerns regarding experiments, the results in the miniImageNet one seem to have impressed some reviewers.

However, all reviewers found the presentation to be inaccurate in more than one points. R1 points out to "issues with presentation" for the hierarchical Bayes motivation, R2 mentions that the motivation and derivation in Section 2 is "misleading" and R3 talks about "short presentation shortcomings".

R3 also raises important concerns about correctness of the derivation. The authors have replied to the correctness critique by explaining that the paper has been proofread by strong mathematicians, however they do not specifically rebut R3's points. The authors requested R3 to more specifically point to the location of the error, however it seems that R3 had already explained in a very detailed manner the source of the concern, including detailed equations.

There have been other raised issues, such as concerns about experimental evaluation. However, the reviewers' almost complete agreement in the presentation issue is a clear signal that this paper needs to be substantially re-worked.